# Assessment of Wayfinding Performance in Complex Healthcare Facilities: A Conceptual Framework

**Ammar Al-Sharaa** [1,*]**, Mastura Adam** [1,*]**, Amer Siddiq Amer Nordin** [2,3]**, Riyadh Mundher** [4] **and Ameer Alhasan** [5]

1 Department of Architecture, Faculty of Built Environment, University of Malaya, Kuala Lumpur 50603, Malaysia
2 Centre on Addiction Sciences (UMCAS), University of Malaya, Kuala Lumpur 50603, Malaysia
3 Department of Psychological Medicine, Faculty of Medicine, University of Malaya, Kuala Lumpur 50603, Malaysia
4 Department of Landscape Architecture, Faculty of Design and Architecture, Universiti Putra Malaysia, Serdang 43400, Malaysia
5 Department of Computer Techniques Engineering, Dijlah University College, Baghdad 00964, Iraq
* Correspondence: ammoratawama@gmail.com (A.A.-S.); mastura@um.edu.my (M.A.)

**Abstract:** Wayfinding is considered to be one of the most demanding challenges to be performed by hospitals' users. Wayfinding has been an interest among researchers from different fields, such as architecture, interior design, cognitive psychology, and facilities management, to name a few. Previous scholars have highlighted the need for a holistic framework taking into consideration both user and environmental factors. A narrative review of the literature was carried out to understand the full extent of the issue and address the ever-increasing demand for a holistic assessment framework. This article attempts to address the underlying gap by proposing a comprehensive framework that takes into account both facets of the issue through a narrative review of the literature to some of the most prominent research attempts to address the problem of wayfinding in complex healthcare settings. Furthermore, the proposed framework can assist both researchers and practicing professionals by providing a comprehensive understanding of the issue of complex wayfinding as well as of the variables to be investigated in the assessment process.

**Keywords:** wayfinding; pathfinding; navigation; indoor environment; interior environment; healthcare facilities; hospitals; assessment framework

## 1. Introduction

Lynch [1] was the first to name wayfinding as a potential problem and formally defined it in his book *The Image of the City.* Lynch [1] recognized wayfinding as a coordinated usage and delineation of sensory environmental signals. This definition led De Jesus [2] to deduce that wayfinding is, in its essence, a matter of spatial orientation. This concept has been evolving: wayfinding was then regarded as the process of navigating a space with an aim to reach a specific destination [3]; at the present time, a widely accepted and more specific definition of wayfinding can be put together as the process in which a person identifies his current relative spatial location and the knowledge of how he can move through the space towards the desired destination as quickly and efficiently as possible [4]. However, the way in which wayfinding can be defined is highly dependent upon the field in which an investigation is carried out and the setting from which the definition is derived [5]. Yet, different definitions of wayfinding have a commonality of elements, based upon which wayfinding can be described as a destination guided motion [6]. The union of spatial and environmental cognition mentioned by Jamshidi and Pati [7] is the reason why people can conduct a thread of decisions incorporating their cognitive abilities to navigate through both the built and the natural environments. The usage of external

environmental communication means such as maps, signs, or geographic positioning system-based navigation systems (GPS), or the lack thereof, does not undermine the reductive definition [8].

The indoor environments of hospitals are some of the most complex environments to navigate due mainly to the high degree of intersection between functions and activities and the variety of functional goals and environmental concerns [9]. Consequently, the higher levels of functional integration have led to the spaces being arrayed in specific patterns to ensure that the required levels of functionality are achieved [10]. According to Ulrich et al. [11], interiors of healthcare facilities have been designed with assertiveness in order to achieve functional objectives. This practice-oriented tendency may create an environment that dismisses the psychological needs of patients, visitors, and staff members [12]. Ulrich et al. [10] regarded the interior environment of these healthcare facilities as psychologically challenging and stressful to users. In this regard, the emphasis on the role of wayfinding and its effects on patients' physical and psychological states are apparent. Furthermore, healthcare buildings are not static; new linkages are often built to connect newly built annexes to the main building [13]. These buildings should thus be viewed as dynamic entities that grow, shift space within their morphology, and change or alter their topology.

What makes wayfinding especially challenging in healthcare settings compared with other structures such as airports and shopping centers is that the wayfinding process in a healthcare setting is highly purposeful at its core. Therefore, it is considered a highly resolute type of wayfinding [14], while wayfinding in airports, shopping centers, and public parks are regarded as a recreational type of wayfinding [15]. Public perceptions of the role of wayfinding in the promotion of recreational walking routes in greenspace have been investigated whereby cross-sectional surveys reveal that urgency levels differ depending on the Space, and Society. Furthermore, hospital patients are considered users who are cognitively operating at a sub-optimal level, which requires them to pay further attention to the nuances of wayfinding.

Given the importance of wayfinding performance to both users and healthcare institutions, there have been a paucity of research initiatives attempting to formalize a conceptual framework for assessing indoor wayfinding performance in complex healthcare facilities that could act as a theoretical guide for future assessment initiatives. Hence, this study's main aim is to develop a conceptual framework for the assessment of wayfinding performance in healthcare facilities.

## 2. Materials and Methods

This study incorporates a set of keywords to investigate wayfinding in complex healthcare settings. The set of keywords ("wayfinding" OR "way-finding" OR "pathfinding" OR "navigation" AND "hospital" OR "clinic" OR "healthcare" AND "interior environment" OR "spatial layout" OR "interior design" OR "interior architecture") was used to collect the available research on wayfinding in healthcare facilities and its implications on human wayfinding performance. Two search engines were utilized to acquire the research articles: the first was Scopus, while the second was four data bases of EBSCOhost, namely Academic Search Elite, Art & Architecture Complete, E-Journals, and Psychology and Behavioral Sciences Collection.

The inclusion criteria implemented by the authors was for the studies to be quantitative, qualitative, or a mixture of both approaches; the studies had to be written in English with a focus on wayfinding performance. Furthermore, only research articles published between 2017 and 2022 were included in the search for relevant articles. Moreover, only articles with available full text and references were included. Research articles were then screened for relevance on two stages, the first stage being the review of the articles' titles and abstract sections and the second being a review of the articles' full content. The screening was to eliminate search results findings that were irrelevant. Additional resources were also included afterwards by employing a snowball technique. Articles were then

analyzed and thematically grouped. A conceptual framework of wayfinding assessment was then proposed to sum up the theoretical connections explaining the mechanism that effects the process of wayfinding, the factors/variables included, and the approaches incorporated. See Figure 1 for further information on the methodology used to collect and sort the research articles.

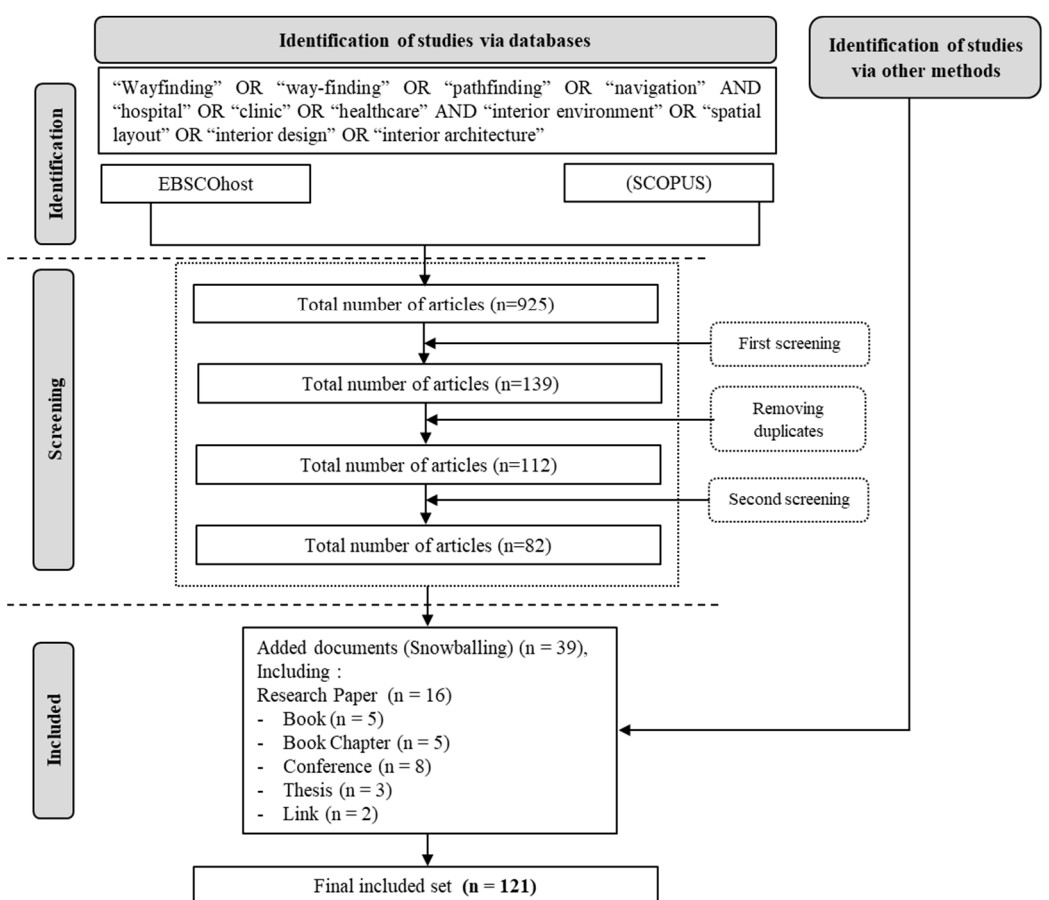

**Figure 1.** Resources collection methodology.

### 3. Results

The initial search results indicated a total of 925 related articles. The first screenings reduced the number of relevant articles to 139; at this stage a total of 27 duplicates were recognized and removed, resulting in a total number of 112 relevant articles. A second screening further reduced the number of total relevant articles to 82. An additional total of 39 research articles were then incorporated through the adapted snowball technique, resulting in a total of 121 articles.

Four thematically linked research domains resembling the domains of the social ecological model (SEM) were recognized in the research articles found within the papers. These domains are listed as per the following: The domain of the physical built environment, the institutional domain, the domain of human cognition, and the social domain. The upcoming sub-sections of this article discuss the most prominent research attempts in each domain that were found to be relevant.

### 4. Assessment of Wayfinding Performance in Healthcare Facilities Based on the (SEM) Model for Human Environment Interaction

The social ecological theory involves four stages of influences in the interaction of people and the environment: individual, social, physical, and policy [16]. SEM aimed to further the understanding of the factors related to people's encounters with the environment by mediating between two polarities: people as the focus of research at the micro level, and the

environment as the focus of research on the macro level. An examination of the interaction of people with a larger environmental context is critical, whereby each level of interaction is complex and may have a consequence for people's experience within the environment, with a tendency to produce a ripple effect through the other layers [17]. The social-ecological model's holistic approach made it a popular choice for investigating the way humans interact with the environment; as a result, the integration of multiple factors that affect the human-environment interaction has become widely accepted. Table 1 summarizes the parameters of the study within the (SEM) framework, and the corresponding factors that may affect people's experiences concerning their engagement with the environment.

**Table 1.** Influencing factors at different domains of the environment.

| Social-Ecological Model Domain | Reference | Highlights | Variables |
|---|---|---|---|
| The individual domain | [18] | Wayfinding induced stress and was correlated with age | • Demographics<br>• Knowledge of the environment<br>• Mental representation of the environment<br>• Prior experience |
| | [19] | Working memory has an effect on wayfinding performance | |
| | [20] | Cognitive map and wayfinding | |
| | [21] | Spatial representation/cognitive mapping | |
| | [22] | Spatial learning and wayfinding | |
| | [23] | Cognitive workload effect on wayfinding | |
| | [24] | Effect of age and fear of confinement on wayfinding | |
| The social domain | [22] | Asynchronous wayfinding in complex environments | • Attitude of people around other people (other users and staff members)<br>• Type of communication (synchronous, asynchronous, etc.)<br>• Availability of information<br>• Clarity of provided information |
| | [25] | Dyad navigation and wayfinding/information synchronicity and wayfinding | |
| | [26] | Geo-crowdsourcing services and collecting accessibility information on the built environment | |
| | [27] | Social role-taking (leading and following) within dyads | |
| The physical environment domain | [9] | Physical properties of hospitals' circulation areas and wayfinding performance | • Location<br>• Orientation<br>• Spatial organization<br>• Legibility<br>• Contrast |
| | [28] | Physical differentiation effect on wayfinding | |
| | [29] | Interior spatial quality effects on wayfinding | |
| | [30] | Physical design elements contributing to wayfinding | |
| | [31] | Information display effects on wayfinding | |
| | [32] | Information display effects on wayfinding behavior | |
| | [33] | Interior organization and layout effects on wayfinding patterns | |
| | [34] | Physical environmental element effects on patients' satisfaction | |
| | [35] | Effects of architectural features on enhancing wayfinding performance | |
| The institutional domain | [34] | Interior design quality evaluation in public inpatient units | • The way-showing devices' properties/Design (i.e., size, color, language, etc.)<br>• Institutional attitude. |
| | [36] | Hospital design guidelines and their potential effects on user satisfaction | |
| | [37] | Institutional wayfinding in complex environments and potential effects on users | |
| | [38] | Organizational outcomes resulting from controlling design variables | |

### 4.1. Models of Wayfinding as a Function of Human Cognition

The association of both environmental and spatial cognition in enabling people's decision making by empowering their cognitive abilities to navigate through the built or natural environment has been widely recognized [7]. However, the occurrence of

wayfinding cognitive processes is not strictly subject to the use of external representations of the environment such as maps, signs, or GPS systems [39]. This implies that wayfinding is a cognitive process in its essence, which may be the reason why the process is complex by nature even though it may seem like a simple process [40].

Wayfinding can be understood as a combination of three interlinked processes: first is the decision-making process, whereby an action plan is developed; it is followed by the process of decision execution, whereby the action plan will be translated into a set of coordinated behaviors; and lastly information processing, which includes the environmental perception and cognition sub-processes that are responsible for the basis of information of the two decision-related processes [7].

Modern empirical research found in the literature has proposed that the process of wayfinding comprises a set of different cognitive processes such as problem solving, decision making, decision execution, and behaviors [21]. A list of theoretical propositions that helped in coining our understanding of those cognitive functions is given in the next upcoming sub-sections, see Table 2.

**Table 2.** Cognitive factors and effects on hospitals' wayfinding performance.

| Factor Classification | References | Remarks |
|---|---|---|
| Demographic factors | [18] | • Wayfinding-induced stress and its correlation with the age of wayfarers |
| | [19] | • Working memory has an effect on wayfinding performance |
| | [41] | • Effect of age and fear of confinement on wayfinding |
| | [42] [43] [44] | • Gender effects on wayfinding performance |
| | [37] | • Level of education effects on wayfinding performance<br>• Native language effects on wayfinding performance |
| Spatial knowledge | [22] | • Spatial learning association with wayfinding performance |
| | [23] | • Cognitive workload effect on wayfinding |
| | [45] | • Familiarity effects on wayfinding performance |
| Mental representation of the environment | [20] | • Cognitive map and wayfinding |
| | [21] | • The association of spatial representation/cognitive mapping on wayfinding performance |

### 4.1.1. Spatial Perception

Theories of Human spatial perception can be described as a group of theories of perception that attempted to explain the mechanisms in which the perception of sensory information occurs; these theories also touched on the mechanisms involved in the processes of information acquisition [21]. Falling under this category of theories are Gestalt theory, the theory of direct perception, template theory, and constructivist theory [7]. Gestalt psychology proposed a set of principles that depict the perception mechanisms of elements by grouping them (Lu and Pesarakli [46]), namely figure-ground, proximity, similarity, continuity, closure, and symmetry. The theory of direct perception argues that our perceptual system can make use of contextual information [47]. It has been described as a theory that relies on data-driven cognitive processing, indicating that information perceived by

our sensory input is satisfactory for perception; it interprets our perception and decision-making by a simple input-processing-output scheme. According to the theory of direct perception, there are no "higher-level" cognitive processes required. This indicates that the environmental cues are satisfactory for environmental perception, which means that our prior knowledge does not account for much of our environmental perception. It was mostly referred to in cases where the role of visual information in navigation was investigated [4].

Perceptual cycle theory emphasized the importance of retrieving information; according to this theory, one collects targeted information from the environment based on the person's prior experiences, then the newly collected information is stored and retrieved, affecting the next cycle of information collection. According to perceptual cycle theory, a person is attempting to coordinate sensory information with a mental template called "schemata" in order to discern patterns in the environment [48]. In this approach to understanding wayfinding, one is constructing concepts; hence, it has been labeled as the constructivist theory [49].

### 4.1.2. Development of Spatial Knowledge

There has been a class of theoretical propositions attempting to explain the stages in which spatial knowledge is developed. Two theories are considered the standouts in this regard, namely Piaget's theory, which investigated spatial development processes in children [50], and the theory of spatial knowledge acquisition by Siegel and White, which builds upon Piaget's theory to examine the same processes among adults especially in an unfamiliar environment [51]. Piaget's theory was mainly concerned with spatial knowledge as a means to understand the mechanisms in which this ability develops during one's lifetime. This theory states that humans possess progressive levels of spatial cognitive abilities. Subsequently, the theory of spatial knowledge acquisition suggested a more general model explaining the mechanisms of acquiring spatial knowledge that applies to subjects of all ages. The theory suggests that the progression of environmental knowledge development in adults experiencing an environment for the first time is like that of children [52].

### 4.1.3. Mental Representation of Spatial Knowledge

The two important concepts in the theoretical propositions that addressed the mental representation of spatial knowledge are cognitive map and cognitive mapping. These theoretical models help to understand how the human mind represents spatial information [53]. Several studies attempted to simulate formation and the representation of the cognitive map by using the Component Process Model (CPM) and the Neural Network Model (NNM) [54]. The concept of cognitive maps argued people's reliance on cognitive maps to navigate through the environment [55]. Cognitive maps are mental representations of the environment whereby Euclidean information, spatial layouts, and affordances of the environment can be cognitively portrayed [56]. Several studies have investigated people's behavior in large-scale environments and have argued that wayfinding behavior can be better explained by investigating cognitive maps [47]. Lynch [1] identified five environmental factors that people acquire to compose a cognitive image: paths, edges, districts, nodes, and landmarks. The concept of Cognitive mapping made use of the "neural network" concept as an analogy to explain the process of cognitive mapping [57]. The cognitive mapping theoretical proposition states that more mental associations are formed between certain locations as one's experience increases within an environment [58]. Numerous mental associations may exist which can be interpreted as a network of associations; the network grows as the person's experience increases in an environment, which in its turn adds up to the spatial knowledge of the person [59].

Cognitive mapping is defined generally as the process of acquiring information from the environment and categorizing it mentally according to its relevance [60]. Jamshidi and Pati [7] distinguished two general approaches in cognitive mapping, the first being the neural network model (NNM) Chang and Leung [61], and the second being connectome-based

predictive modelling (CPM) [62]. The CPM model is primarily based upon the Information Processing Theory (IPT) [63] in which the human cognitive system is considered as a computation unit. According to the CPM model, spatial information is considered a cognitive input, then human cognition will inspire changes to the information in accordance with a pre-determined set of rules, after which the spatial learning process will subsequently occur.

### 4.1.4. Spatial Cognition

A set of attempts focused on explaining the process of active wayfinding based on an input such as spatial data to an output such as actions [64]. Two main theoretical structures have emerged under this category, the Information Processing Theory (IPT) and theories of problem-solving [65]. According to IPT there are similarities between the way human beings register information and the way computers receive data input [66]. The theory goes beyond, viewing the act of thinking as analogous to a computer program, and human memory or the capacity to store information as analogous to the amount of information stored on a computing device that can be measured by bits of information. Furthermore, the theory considers forgetting some information to be similar to the process of actively deleting information from a computing device, and the process of recalling previously registered information as a similar process in principle to using an information search function. The theory also compares strategizing to using computer scheduling tools, and finally the process of decision making to computer output [66]. Information processing scholars consider the structure of human cognition comparable to the architecture of computers, and often refer to it as cognitive architecture [67]. Theorists in this subject communicated their findings which illustrated the information flow by utilizing representations of the human brain as an information processing apparatus [68]. Many have expressed the information flow by illustrating the steps of the flow in the form of flow diagrams, often referred to as models [69]. See Figure 2 for a diagram illustrating the information processing mechanisms that interpret the cognitive information processing as a multi-level processing system.

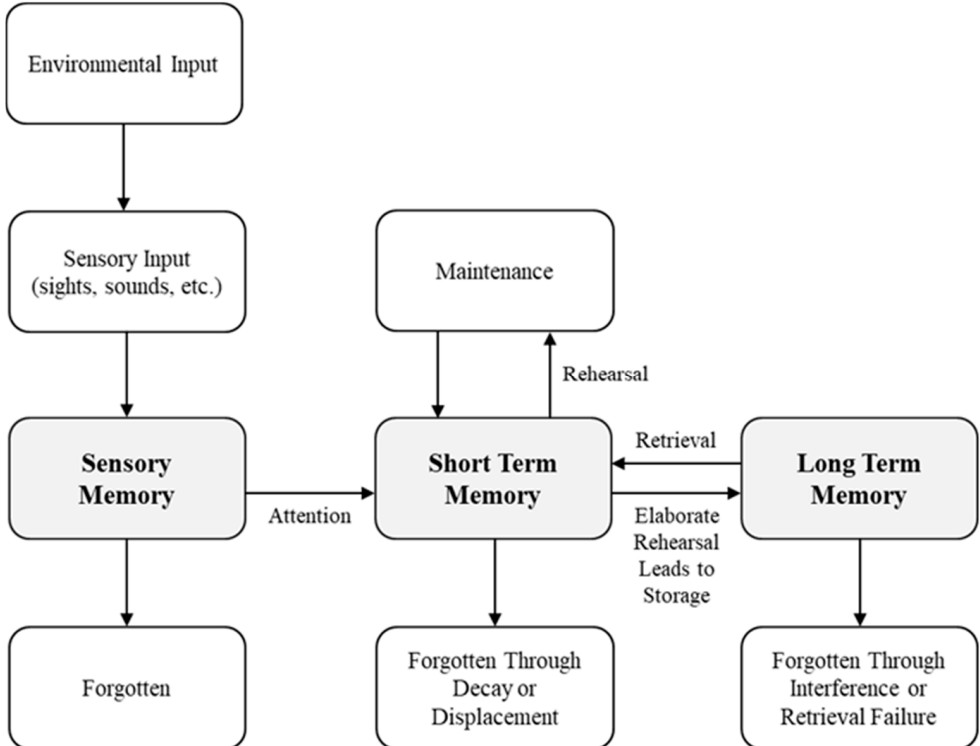

**Figure 2.** Multi-level model of information processing [69].

A model called Test-Operate-Test-Exit (TOTE) attempted to explain information processing involved in the process of problem-solving, which is described as incorporating

a feedback loop procedure in order to solve problems [70]. In this model, one examines the achievement of goals, and in cases where goals have not been achieved a reaction in accordance with the feedback is occurring, upon which examination starts again [71]. This cyclical process continues until objectives are met. Similarly, a successful wayfinding procedure is based on a continuous examination and feedback processes [72]. The examination procedure can be seen in wayfinding-related problem-solving strategies such as the process of rout Control, in which wayfinding action continues if no alternative route is deemed more affordable [21].

### 4.1.5. The Evolutionary Theoretical Proposition for Spatial Sex Differences

The hunter gatherer theory is a theory highlighting spatial differences between sexes. These differences were attributed to various empirical evidence showing differences in the performance of spatial tasks between genders [73]. Several research articles argued that an evolutionary theory can explain the root of these differences [7]. The argument of an evolutionary origin was developed based on the fact that males were engaged in hunting activities more frequently than their female counterparts; on the other hand, females were engaged more in foraging. This led to the variation in spatial development and capabilities which were more suitable for the tasks that they were engaged in. Accordingly, males needed to develop abilities such as mental rotation, which is essential to successful hunting. The development of mental rotation can facilitate map reading and maze learning [42]. Hence, males outperform females in these kinds of spatial tasks [74]. On the other hand, successful foraging needed the ability to memorize objects and their location for further inspections [75]. That explains the superiority of females in recalling objects and their locations compared to males in several studies [43]. Moreover, several empirical studies suggest that spatial abilities such as mental rotation and sense of direction play an important role in explaining gender differences in wayfinding [76]. Therefore, the hunter-gatherer theory attempts to explain the reason spatial sex differences exist.

### 4.2. The Social Domain in Wayfinding

Despite the numerous studies investigating wayfinding as a primary or a secondary objective, there has been a paucity of research focused on studying wayfinding as a social activity as it has been treated mostly as an issue facing individuals. Similarly, studies of agent-based modeling, in which pedestrian agents are created to simulate pedestrian movements, treated pedestrian movement as movements of single units without any interaction with other simulated agents or groups [77,78]. Further research on the social aspects of wayfinding is in the field of research of verbal communication in wayfinding instructions, which focuses on communicating route directions [79]. Dalton et al. [25] indicated that examining instructions in wayfinding involves studying social wayfinding implicitly due to the involvement of multiple individuals, while mostly focusing on the content of the information supplied in the instructions such as word count, instructions platform, and landmarks. Some research attempts focused on either dyad navigation or the effects of the presence and behavior of other navigators on one's wayfinding performance and behavior [27,80]. This suggests that wayfinding instructions in both forms, direct and indirect, are a type of social wayfinding.

Forlizzi et al. [81] noted the interaction between traveling pairs when using vehicular navigation systems, which yielded a set of findings regarding the design of navigation systems. Another study conducted by Haddington [82] focused on gestures and conversational exchanges among people regarding vehicular navigation. Similarly, He et al. [83] examined pairs of individuals' walking route choices in an unfamiliar urban environment. These research initiatives focused on observing social interactions occurring among co-navigators in a particular setting. Dalton et al. [25] hypothesized that social wayfinding can be investigated further beyond simply co-navigating pairs (or, as researchers describe them, "dyads") and beyond simple synchronous interaction between co-navigating members.

An interesting study of possible social interactions among multiple wayfinding people was conducted by Haghani and Sarvi [84], who examined the effect that social behaviors have while performing a simulated emergency evacuation. The study's results suggested that peoples' emergency exit strategies are heavily affected by both the physical environmental factors and social interactions. However, this paper's sole focus was investigating people's wayfinding behavior in an emergency event, and it therefore may not be predictive about other wayfinding tasks.

Bae [85] indicated that problem-solving and reasoning is primarily a socio-cultural process, found in the communication of groups of individuals and artifacts generated by previous group members. Hutchins [86] also found that group cognition could vary from the cognition of the agents taking part in performing the task within the group and is different from the total sum of the contributions made by the individuals forming the group.

### 4.2.1. Types of Social Wayfinding

Dalton et al. [25] proposed a classification of social wayfinding into two types, Strong and Weak social wayfinding. Strong and Weak social wayfinding were defined according to the embedded "degree of intentionality" in the process of exchanging information. In the case of Strong social wayfinding, it was defined as an intentional exchange of information about the wayfinding process between "co-navigators", where more than a single individual exchanges information regarding the location or the route choice. Alternatively, Weak social wayfinding was defined by the unintentional communication between the actors which means that there is no occurring co-navigation. Weak social wayfinding takes place when individuals unintentionally communicate information regarding the location or the choice of rout. In this type of social wayfinding, cues are generally created by the individual without the intention of providing them, as a by-product of their own navigation. Strong social wayfinding generally involves direct communication between senders and recipients, most often between individuals in proximity. On the contrary, Weak social wayfinding often occurs remotely, within the extent of sensory access. Wayfinding was also classified, according to the timeframe of the information exchange, into synchronous and asynchronous. Therefore, each of the types can be divided according to the synchronicity of communication among individuals, resulting in a total of four types of social wayfinding, see Table 3.

### Synchronous Social Wayfinding

Synchronous social wayfinding takes place when an influence from others occurs during the navigation process. In this case, one makes their decisions with accordance to an intentionally generated external input. The contribution of an external information source is direct and intentional. When conducted by another co-navigator within proximity, by incorporating verbal and/or gestural communication, it is called Strong synchronous wayfinding. When others navigate alongside each other, it is safe to assume prior knowledge amongst these people. However, information inquiry and exchange can be a form of social interaction among people. In contrast to Strong social wayfinding, other people can influence wayfinding decision-making in indirect and unintentional ways, which is referred to as Weak social wayfinding. When this Weak influence expresses itself during the actual travel of a navigating person, it is not only Weak but also Synchronous. This type of wayfinding can be best represented in studies in which participants were finding their destinations within a space by incorporating verbal communication as a means of communication [88–90] or by mimicking other people's wayfinding behavior [41,87].

### Asynchronous Social Wayfinding

According to Dalton et al. [25], social wayfinding can take place without synchronicity. In these cases, information exchange regarding wayfinding decisions takes place at a time prior to the travel in the form of given instructions. These instructions can take various forms, such as written, spoken, or gestured. Another form of achieving asynchronous

social wayfinding can take place by the usage of maps, which are a more common mode of instruction delivery in asynchronous wayfinding than they are in synchronous wayfinding. It is worth noting that in both synchronous and asynchronous wayfinding, supplying information by a non-traveler is considered a less engaging system than Strong wayfinding. The process of supplying instructions to other navigators is usually reliant on one person assisting another without accompanying that second person during the travel.

**Table 3.** Types of social wayfinding and related studies.

| Factor | Type | References | | Remarks |
|---|---|---|---|---|
| Communication | Strong | [22] | • | Asynchronous strong wayfinding in complex environments |
| | | [25] | • | Dyad navigation and wayfinding/information synchronicity and wayfinding |
| | | [27] | • | Social role-taking (leading and following) within dyads |
| | Weak | [25] | • | Dyad navigation and wayfinding/information synchronicity and wayfinding |
| | | [26] | • | Geo-crowdsourcing services and collecting accessibility information on the built environment |
| | | [87] | • | Occupants' behavior in an emergency scenario and the effect of the crowd |
| Synchronicity | Synchronous | [88] | • | Asking people passing by for verbal assistance |
| | | [89] | • | The effect of verbal memory on wayfinding familiarity |
| | | [90] | • | The use of mobile applications in wayfinding to assist the visually impaired in improving wayfinding and safety |
| | Asynchronous | [91] | • | The use of social media digital cues as a wayfinding aid |
| | | [92] | • | The use of mobile digital technology as a navigation aid |

*4.3. Wayfinding and the Physical Built Environment*

Wayfinding can be considered an important aspect of environmental spatial quality. Colenberg et al. [93] defined the spatial quality of environmental interiors by the structure and enclosures formed by the architectural elements which comprise the environment, such as floors, ceilings, walls, windows, and vertical movement bridging elements such as stairwells and elevators. Interior elements of the environment can serve both visual and functional purposes by incorporating elements such as materials' properties, construction elements' properties, and technology. Visible elements of interior environments conform interior spaces into a habitable and functional space. Similarly, the American Society of Interior Design (ASID) (2008) [94] considers interior design, in its core, a functional process that can enhance living quality as well as being culturally influential. The (2008) published interior design manual goes further in detailing dimensions of interior design quality including outcomes of productivity, health, safety, and wellbeing of the users. According to a study conducted by Abu Samah [29], components of spatial quality in healthcare settings can be broken down to its principal components, namely technical, functional, and aesthetic. It is worth noting that there was an overlap between components of spatial quality and components that were investigated in the context of wayfinding studies, see Table 4. The next three subsections list the elements that have been mentioned in the literature whose effect on wayfinding performance was measured directly or indirectly.

**Table 4.** Literature on the components of the interior environment that were found to be affecting wayfinding performance in hospital buildings through affecting perceived spatial quality.

| Component(s) Classification | Factors/Variables | References | Remarks |
|---|---|---|---|
| Technical | 1. Safety<br>2. Lighting | [95] | • Appropriate lighting promoted indoor wayfinding |
| | | [96] | • People feel safer in a uniformly lit environment<br>• Separated light zones enhance wayfinding, as they can be read together as a coherent pattern |
| | | [97] | • Color temperature does have a significant effect on hesitation |
| | | [98] | • Lighting's effects on perceived safety |
| Functional | 1. Accessibility/permeability<br>2. Spatial layout<br>3. Information display | [23] | • Information formatting's effects on wayfinding performance |
| | | [74] | • The effects of map design on wayfinding performance |
| | | [99] | • Users' permeability in two different typologies |
| | | [100] | • Spatial layout's effects on wayfinding and the level of social contact amongst individuals |
| | | [101] | • Spatial layout complexity and similarity effects wayfinding performance |
| | | [102] | • Spatial layout typology and wayfinding performance |
| Aesthetic | 1. Colors<br>2. Materials and textures | [103] | • Color helps in spatial identification<br>• Color conspicuity and information comprehensibility are hard to consolidate simultaneously |
| | | [104] | • The texture of the environment's finishing materials can affect navigators' safety |

### 4.3.1. Technical Components' Effect on Wayfinding Performance

Abu Samah [29] classified lighting, thermal comfort, air quality, noise, and safety as technical components. Two elements, air quality and thermal comfort, have no obvious direct effect on the process of wayfinding. Meanwhile it is widely agreed upon that wayfinding and other ergonomic interior design elements might reduce the number of falling staff members and the number of injuries occurring, and could also reduce potential violent actions [105]. Alternatively, two other technical elements, light and noise, have been discussed by the studies concerning wayfinding and indoor navigation extensively and are considered to have a direct significant effect on wayfinding performance. Studies showed that subjects are more likely to have a bias towards the brighter lit paths as opposed to the darker ones. Higher noise level is shown to have a potential negative effect on certain patient outcomes [106,107]. There is also some evidence that staff personnel might experience higher stress levels caused by higher sound levels in patient units [108].

### 4.3.2. Functional Components' Effect on Wayfinding Performance

Wayfinding was discussed as a functional aspect of the hospital's design that contributes to its perceived spatial quality. Functional aspects of spatial quality were divided into spatial planning, accessibility, wayfinding, and furniture. It is our understanding that accessibility and wayfinding are both characteristics of the spatial layout that vary from one spatial layout to another, accentuated by an indoor purposefully designed and scattered items serving as wayfinding cues and features to achieve spatial differentiation, especially in cases where the interior environment lacks the means in which it could differentiate its parts from one another without causing confusion and cognitive load. Architectural wayfinding design was defined by Hunter [109] as the function that addresses the built components, including spatial planning, articulation of form-giving features, circulation systems, and environmental communication. Good wayfinding systems should go beyond mere signage and the use of color codes to differentiate various hospital areas; this emphasizes the role of a good wayfinding system on both a user level and an organizational level [21]. Spatial layout is directly correlated to all three environmental physical properties, namely appearance differentiation, visual accessibility, and complexity of the building's layout [110]. Visual access is the environmental extent that allows human vision to observe features and objects [110], while accessibility is usually correlated to the sequence of activities. Furthermore, accessibility is related to users' movement through the building environment from their origin of movement to their desired destination. Accessibility of physically disabled users is usually the focus when inclusive design is an objective of the institution [111]. Lighting is another functional aspect of spatial quality showing a direct effect on wayfinding performance [34]. Experimental evidence indicates that users are more likely to favor a well-illuminated environment when selecting routs. There are two types of studies that can be found in the published literature: the first is focused on indoor lighting in hospitals generally and its effects on wayfinding performance [80,95], while the second type's focus is primarily on natural lighting and its potential for psychological relief [112,113].

### 4.3.3. Aesthetic Components' Effect on Wayfinding Performance

Color can affect the way in which direction signage and orientation systems are perceived, which contributes to aspects of safety, efficiency, wellbeing, and spatial verification [114,115]. Multiple research studies have mentioned the importance of color in creating a pleasant and comfortable feeling towards the environment [116,117]. Contrasting colors and intensities may aid in the process of discernment of different spatial elements for the purpose of definition, separating different areas, showing directions, marking floor levels, signaling intersections, and indicating destinations. Previous research also indicated that color should be used carefully and recommended that only a limited number of contrasting colors be used, due to the possibility of users, especially those undergoing highly stressful circumstances, not distinguishing and memorizing subtle nuances [103,118]. Materials and finishes of the interior environment also seem to influence wayfinding: literatures suggest an effect on safety, specifically avoiding slippery flooring which can lead to injuries [104,119].

### 4.4. Institutional Influences on Wayfinding in Healthcare Facilities

Empirical studies indicated that complex medical facilities with excessive institutional environments have triggered several undesirable outcomes for their users, with wayfinding being one of them [5,37]. Other research focused on identifying these adverse outcomes, such as the study conducted by Jiang and Verderber [9], which highlighted the potential resulting outcomes of these environments commonly described as challenging, among which causes of high levels of stress and anxiety, loss of perceived control, and insufficient accessibility to positive distractions were described as the lack of meaningful interaction with nature. A number of published research articles mentioned the institutional role in wayfinding at healthcare facilities [29,34]. An example of how institutional nuance can

make a slight difference due to difference in priorities was the example of teaching hospitals, which are under the institutional jurisdiction and management of the Ministry of Education (MoE) [5,36]. Teaching hospitals serve their role as constituents of a Healthcare System with a slightly nuanced set of priorities geared towards learning and teaching. On the contrary, privately owned and operated healthcare facilities are not fully integrated with the national healthcare system and operate under their own accordance, while following the rules and regulations set by their respective governing bodies [120,121]. This indicates that healthcare facilities are mandated by the regulations of their own respective governing bodies.

Institutions do have some level of room to innovate and nuance themselves to stand out from the competition. These institutional decisions can potentially affect the experience of patients and families as well as other institutional outcomes [122,123]. Wayfinding researches in highly institutionalized environments have been conducted in multiple settings such as libraries [124], university campuses [37], airport terminals [125], train stations (112), and hospitals [126]. These studies assume that the spatial layout is designed in a pre-construction phase and carried out to operation as designers intended, and that once facilities are set to a location they remain there. However, upgrading facilities and changes in recommended practices are frequent enough to consider spatial reassignment every now and then, which in its turn can influence wayfinding via affecting the environment. Institutions, in most cases, like to be involved in the processes of decision making especially when the decisions may lead to institutional outcomes such as the outcomes mentioned by Ulrich et al. [38], who highlighted a set of thirteen potential institutional outcomes resulting from the implementation of evidence-based design (EBD) practices. Wayfinding problems can be addressed effectively by integrating improvements to the physical environment with organizational and operational changes [127].

## 5. Wayfinding Assessment Approaches

Wayfinding assessment approaches can be sorted into three categories based on their approach (See Figure 3). Studies concerned with human cognitive abilities, and the variations in wayfinding capabilities among different demographic groups, were in general focused on how a physical or cognitive impairment can affect the overall performance and how these effects can be mitigated; these studies showed promising suggestions for how the hospital's interior environment can play a crucial role in achieving inclusivity.

**Perceptual Approaches**
- Can be either qualitative or quantitative
- The focus is users' spatial perception and cognitive processes related to wayfinding

**Hybrid Approaches**
- Using techniques such as VR, AR, VE to assess environmental attributes of physical structures through measurement of user performance.
- Using mobile applications
- Using crowdsourced data

**Spatial Approaches**
- Wayfinding performance is a physical environmental attribute whereby spatial properties are the focus
- Optimization of wayfinding time & distance is the goal.

**Figure 3.** Research approaches in the field of wayfinding.

En and bin Bebit [128] conducted a study focused on analyzing reactions to signage design within a healthcare facility, whereby a qualitative approach by adopting an interview was suggested as an examination technique. Moreover, a study conducted by Zijlstra et al. [129] focused on simulated physical ageing indicated that rout complexity can affect aging people negatively when a wayfinding experiment was conducted to simulate

the elderly. Other Studies have focused on the effect of cognitive impairments such as dementia and Alzheimer's on wayfinding performance in healthcare facilities [9]. Moreover, there has been an interest in the physical environment as the basis in which the lived or perceived space was cognitively formed; therefore, the focus was steered towards the physical elements of the environment. In a comprehensive multi-method non-experimental, qualitative, exploratory design study, Pati et al. [30] found that physical design elements contributing to wayfinding include signs, architectural features, maps, interior elements (artwork, display boards, information counters, etc.), functional clusters, interior elements pairing, structural elements, and furniture.

Conversely, an emergent pattern in recent studies was to try to establish a quantitative approach to assessing wayfinding performance while still encompassing user experiences as a factor. The usage of visual reality models [30] and eye tracking [46] offer a more transferable wayfinding assessment data that were not possible without the usage of modern-day technologies. The use of these emerging technologies has enabled recent researches to bridge the gap between studies focusing solely on the user experience or on the physical environment [24,42].

Furthermore, studies such as Vizzari et al. and Khasraghi [77,78] were focused on developing methods of quantizing human spatial behavior based on visual input. In addition, a set of spatial measures that are either visual, such as Isovist measurands [115], or topological, such as spatial syntactical measurands [130], were suggested to be a predictor of wayfinding performance.

This taxonomy of approaches is consistent with the one proposed by [131,132]; although it was not wayfinding-specific, the principles of human-environmental interaction are similar.

## 6. Conceptual Framework for the Assessment of Wayfinding Performance in Complex Healthcare Facilities

The primary investigation of literature led to the realization that wayfinding was investigated primarily from within one of four domains that represents a set of thematically linked factors. This study proposes a conceptual framework for wayfinding performance assessment that considers all four contributing domains of wayfinding: the domain of human cognition, the domain of the physical built environment, the social domain, and the institutional domain.

The four aforementioned domains contain a set of variables that have been the primary target of investigation in the available literature on wayfinding. The proposed framework highlights the variables that are considered the primary predictors of wayfinding performance in general and in healthcare facilities specifically.

The wayfinding task is governed by both the environmental and the institutional factors by means of environmental and institutional constraints. The environmental constraints are interpreted as the sum of the constraints that are imposed by the environment on the user when the user starts his wayfinding journey in the environment. The subsequent assessment approaches are also highlighted in the proposed framework, as illustrated in Figure 4.

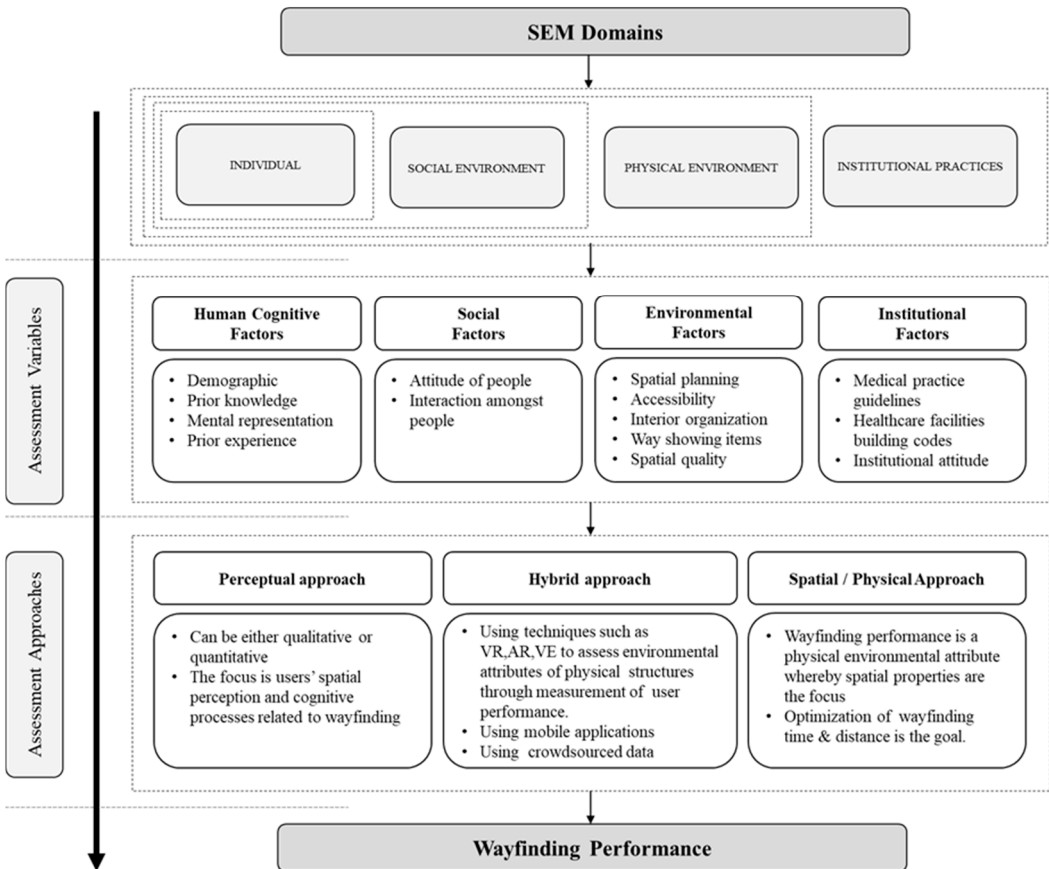

**Figure 4.** Conceptual framework for wayfinding performance assessment.

## 7. Limitations and Future Work

Despite its findings, this study has several limitations. Firstly, the study employed a set of keywords to acquire research articles that are used to formulate the understanding of terms and mechanisms involved in indoor wayfinding in healthcare facilities; the incorporation of other keywords might reveal another dimension of wayfinding performance in healthcare facilities outside of what is discussed in this paper.

Secondly, this study has employed the search results of only two engines, namely Scopus and EBSCOhost; incorporating research articles from other search engines such as web of science (WOS) could further enrich the findings of the study and may have offered a set of articles offering a different perspective from the findings discussed in this research.

Thirdly, this study included research articles published between 2017 and 2022, and while the snowball technique managed to include some articles out of this timeframe, conducting a research articles survey that incorporates research articles from a longer timeframe could offer a perspective on the linear progression of research in wayfinding in general as well as wayfinding research in healthcare facilities.

Furthermore, this study discusses significant aspects of the assessment of wayfinding performance in healthcare facilities; however, a quantitative analysis is not included. Future research initiatives are encouraged to further investigate the quantitative aspects of published research studies within the subject matter.

## 8. Conclusions

Wayfinding in complex healthcare facilities can be a highly taxing process on both patients and visitors. This challenging process can have multiple physical and psychological implications which result indirectly in potential institutional adverse outcomes such as monetary losses, energy waste, and distraction of staff members, as well as potentially affecting the institution's image. This manuscript's main aim is to conduct a review of the

available literature on the assessment of wayfinding performance in healthcare facilities to then suggest an assessment conceptual framework. This study also provides a generic wayfinding definition that is based on the theoretical background discussed in this paper. The manuscript offers a set of potential implications for both theoretical and practical fronts by presenting a generic definition for wayfinding which can enrich the current understanding of the wayfinding process, as well as the conceptual assessment framework which can form the basis for future wayfinding performance assessment initiatives. Future research initiatives focusing on the assessment of wayfinding performance in complex healthcare facilities can utilize this conceptual framework in terms of identifying the factors and variables involved in wayfinding in complex healthcare facilities.

**Author Contributions:** Conceptualization, A.A.-S. and M.A.; data review, A.A.-S., A.A. and M.A.; writing—original draft preparation, A.A.-S. and A.S.A.N.; review and editing, A.A.-S., M.A., A.S.A.N., R.M., and A.A.; visualization, R.M. and A.A.; supervision, M.A. and A.S.A.N. All authors have read and agreed to the published version of the manuscript.

**Funding:** This research received no external funding.

**Institutional Review Board Statement:** Not applicable.

**Informed Consent Statement:** Not applicable.

**Data Availability Statement:** Not applicable.

**Conflicts of Interest:** The authors declare no conflict of interest.

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
