# Peer review of "Assessment of Wayfinding Performance in Complex Healthcare Facilities: A Conceptual Framework"

_sustainability, doi:10.3390/su142416581_

Round 1

Reviewer 1 Report

The authors presented a literature review aimed to address the problem of wayfinding in complex healthcare settings.

While the topic is relevant some changes should be carried out before considering the manuscript as a potential work for publication.

First, the abstract should be written in the format established in the author's guidelines.

The introduction section is well written, nevertheless, it is not easy to understand and justify a new review of the literature. What are the questions that are you trying to solve?

While the narrative reviews do not necessarily require a methods section, it is strongly desirable to write some paragraphs explaining how the authors reached the references.

Tables 2 and 3 are highly valuable, but the "remarks" may be explained in more detail. 

The references should be written in the same line that the paraghraf (i.e. line 252 starts with the number 83, which corresponds to the last line). Also, review the guidelines of the journal for citation.

Figure 2. The authors show the research approaches in the field of wayfinding. It seems that it is the same proposed by Mundher for urban environments and adapting it. Please clarify.

Please, clarify what was the role of ASAN and AA-H. Check if those authors fullfil the criteria (i.e. https://www.icmje.org/recommendations/browse/roles-and-responsibilities/defining-the-role-of-authors-and-contributors.html) 

Please, add a paragraph with the limitations, questions and further research that may be formulated from this review.

Reviewer 2 Report

The manuscript has a lot of merit in its conformation and relevance in the study area, but it requires adjustment in its sequence, grammar, and adjusting with respect to the author's rules (for example, the references are not in accordance with what is required by the magazine).
On the other hand, authors should review their review process and it will be worth a lot if they can adhere to the PRISMA review guidelines (The PRISMA 2020 statement: an updated guideline for reporting systematic reviews
BMJ 2021; 372 doi: https://doi.org/10.1136/bmj.n71 (Published 29 March 2021).
In this sense, the information presented in tables 1-4 could be expressed in percentages to make comparisons in their different groups and include the tables with their specifications as supplementary material. Suggestions that would allow showing patterns in the results obtained and analyzed in the reviewed articles.

Finally, it is necessary for the authors to carry out an adequate review of the language because there are sections that make it difficult to read and understand.

Round 2

Reviewer 1 Report

The authors have satisfactorily addressed the suggestions and changes suggested by the referees.

Congratulations!

Reviewer 2 Report

The adjustments and changes applied to the new version of the manuscript are appropriate. Also, when reviewing the document, they attended to the indications and incorporated relevant information that reviewer 2 delimited.